# Interactions at the Oviposition Scar: Molecular and Metabolic Insights into *Elaeagnus angustifolia*’s Resistance Response to *Anoplophora glabripennis*

**DOI:** 10.3390/ijms25179504

**Published:** 2024-08-31

**Authors:** Chengcheng Li, Jiahe Pei, Lixiang Wang, Yi Tian, Lili Ren, Youqing Luo

**Affiliations:** 1Beijing Key Laboratory for Forest Pest Control, Beijing Forestry University, Beijing 100083, China; lcc311@foxmail.com (C.L.); jiahepei@bjfu.edu.cn (J.P.); ty200609@sina.com (Y.T.); 2Biocontrol Engineering Laboratory of Crop Diseases and Pests of Gansu Province, College of Plant Protection, Gansu Agricultural University, Lanzhou 730070, China; wanglx@gsau.edu.cn

**Keywords:** Asian long-horned beetle, dead-end trap tree, Russian olive tree gum, starch and sucrose metabolism

## Abstract

The Russian olive (*Elaeagnus angustifolia*), which functions as a “dead-end trap tree” for the Asian long-horned beetle (*Anoplophora glabripennis*) in mixed plantations, can successfully attract Asian long-horned beetles for oviposition and subsequently kill the eggs by gum. This study aimed to investigate gum secretion differences by comparing molecular and metabolic features across three conditions—an oviposition scar, a mechanical scar, and a healthy branch—using high-performance liquid chromatography and high-throughput RNA sequencing methods. Our findings indicated that the gum mass secreted by an oviposition scar was 1.65 times greater than that secreted by a mechanical scar. Significant differences in gene expression and metabolism were observed among the three comparison groups. A Kyoto Encyclopedia of Genes and Genomes annotation and enrichment analysis showed that an oviposition scar significantly affected starch and sucrose metabolism, leading to the discovery of 52 differentially expressed genes and 7 differentially accumulated metabolites. A network interaction analysis of differentially expressed metabolites and genes showed that *EaSUS1*, *EaYfcE1*, and *EaPGM1* regulate sucrose, uridine diphosphate glucose, α-D-glucose-1P, and D-glucose-6P. Although the polysaccharide content in the OSs was 2.22 times higher than that in the MSs, the sucrose content was lower. The results indicated that the Asian long-horned beetle causes Russian olive sucrose degradation and D-glucose-6P formation. Therefore, we hypothesized that damage caused by the Asian long-horned beetle could enhance tree gum secretions through hydrolyzed sucrose and stimulate the Russian olive’s specific immune response. Our study focused on the first pair of a dead-end trap tree and an invasive borer pest in forestry, potentially offering valuable insights into the ecological self-regulation of Asian long-horned beetle outbreaks.

## 1. Introduction

The Asian long-horned beetle (ALB), *Anoplophora glabripennis* (Motschulsky, 1854) (Coleoptera: Cerambycidae), is native to eastern Asia and poses a significant threat to hardwood trees in North America and Europe. ALBs inflict damage to various broad-leaved trees, including *Populus*, *Salix*, *Betula*, *Acer*, and *Ulmus* trees, primarily through larval feeding in branches. The spread of the ALB across China, driven by seedling transportation, extended to Harbin City in Heilongjiang Province by 1959, to Ili City in the Xinjiang Uygur Autonomous Region by 1999, and to Lhasa and Shigatse in Tibet by 2002 [1,2,3,4]. Furthermore, the ALB was introduced to North America, starting in New York, USA, in 1996, and has spread to several European countries, including France in 2003 and Germany in 2004, through global trade [5,6,7].

The Russian olive, *Elaeagnus angustifolia* Linn. (Myrtiflorae: Elaeagnaceae), possesses significant ecological value and plays a crucial role as a windbreak and in sand fixation, soil and water conservation, vegetation restoration, and afforestation in Asia [8,9]. The Russian olive, which is native to Eastern Europe and Asia, was initially introduced to various regions in the late 1800s for its ornamental value, as a windbreak, for erosion control, and for wildlife management, but it became an invasive species and spread beyond cultivation across the region in the 1950s [10,11]. Adult ALBs can obtain their nutrition and create OSs on the Russian olive’s twigs and branches [12]. The Russian olive offers an ideal environment for ALB adults to nourish and oviposit, yet it inhibits egg hatching by gum secretion at these scars [13,14]. In a previous forest survey, we systematically observed and scientifically described the phenomenon of Russian olives attracting and killing ALBs [15]. The process by which female ALB eggs are killed by the Russian olive is as follows: (1) a female beetle bites an oviposition scar (OS) with its mouthparts on a branch; (2) the female inserts its ovipositor to OS; (3) an egg is deposited within the OS; (4) secretions are produced at the OS to establish a favorable microenvironment for egg incubation and to seal the scar; (5) the Russian olive secretes gum at the OS, gradually occupying the space and encapsulating the egg; and (6) gum emerges externally from the scar, resulting in the demise of the enclosed eggs (Figure 1).

A “lure-kill” relationship has been established between Russian olive and ALB, and this kind of relationship has been utilized in agriculture, called the “dead-end trap plant” method. This model employs specific plant species to attract and eliminate insect pests and has been integrated into agricultural pest management strategies. For instance, the diamondback moth, *Plutella xylostella* (Linnaeus) (Lepidoptera: Plutellidae), is among the most destructive pests affecting cruciferous crops globally. Shelton and Nault (2004) [16] demonstrated that the Gtype of *Barbarea vulgaris* R. Br. var. *arcuata* (Brassicaceae) can attract *P. xylostella* for oviposition, with none of the larvae surviving on this plant, indicating its potential as a dead-end trap for this pest in the field. Shelton et al. (2004) termed this type of plant a “dead-end trap plant” [16]. These plant species attract adult female pests for oviposition while inhibiting pest proliferation. Given these examples, the Russian olive can be regarded as a “dead-end trap tree” of ALBs. Our study focuses on the first pair of dead-end trap trees and invasive borer pests in forestry, potentially offering valuable insights into the ecological self-regulation of ALB outbreaks.

However, research on the fundamental principles, mechanisms, and field applications of “dead-end trap trees” in forestry remains insufficient. This paper discusses and elaborates the concept of a “dead-end trap tree”: a tree that effectively attracts adult insect pests and eliminates their offspring, thereby impeding population continuity.

As previously researched, Russian olive gum, which plays an important lethal role in ALBs, is a secretion exuded from the stem and branches of the tree. It can be used medicinally or as a substitute for Arabic gum [8,17]. The primary component of Russian olive gum is polysaccharide (89%), which consists of 22.7% D-galactose, 41.8% L-arabinose, 8.7% L-rhamnose, 3.5% D-glucuronic acid, and 1.2% D-mannose [18]. Russian olive gum is predominantly secreted from wounds, with prior research showing that the type of wound—whether mechanical or insect-induced—affects both the quantity of gum produced and its secretion rate [8,18,19,20,21]. We have summarized the effects of different factors on Russian olive gum secretion in previous studies (Appendix A).

Plants exhibit differing responses to mechanical scars (MSs) and biological scars, which are linked to molecular mechanisms. Herbivore-associated molecular patterns (HAMPs) and damage-associated molecular patterns (DAMPs) are molecules produced in response to insect predation [22]. For instance, *Spodoptera exigua* induces maize plants to release volatiles that attract parasitic natural enemies of the beet nightshade moth, a response that mechanical damage alone cannot elicit [23]. Plants possess both constitutive and induced defense mechanisms against insect herbivores. Following herbivore injury, plants rapidly activate their induced defenses. Insect species may be directly affected by toxicity or antinutritional factors, or they may experience indirect effects [24,25,26]. Those molecules are recognized by receptor proteins on or within the plant, activating defense signaling pathways that enable the plant to induce systemic resistance, thereby protecting against insect predation and damage. We hypothesize that Russian olive may respond differently to OSs compared to MSs, with gum secretion potentially regulated by defense mechanisms associated with HAMPs.

The objective of our study was to investigate the specific response of gum secretion in Russian olive trees to infestation by the ALB, particularly during the oviposition phase. We aim to analyze the differences in Russian olive gum secretions between the OSs caused by ALBs and the MSs caused by mechanical tools on Russian olives. Our comparative analysis will focus on metabolites and genes, as well as their interactions. This study elucidates the reaction of Russian olive to ALB oviposition scars and establishes a foundational understanding of the molecular processes underlying the lethal effects of Russian olive gum on ALB eggs.

## 2. Results

### 2.1. ALBs’ OS Quantity on the Russian Olive

Out of the 60 Russian olives surveyed at the sample site, 27 did not have OSs. The maximum number of OSs on a single Russian olive was 300, with an average of 33.2 ± 57.2 per tree, indicating significant variability among individual trees (*p* < 0.0001, t = 4.486, df = 59) (Appendix A). Gum production became apparent within 24 h, with an average external weight of 0.49 ± 0.31 g (mean ± SD, n = 30) (Appendix A).

### 2.2. Differences in the Gum Rate and Weight between OSs and MSs in Russian Olive Trees

After 50 days, gum secretion was observed in all MSs, with an average gum block weight of 0.20 ± 0.07 g (mean ± SD, n = 18) (Figure 2). In contrast, the OSs exhibited a secretion rate of 95.08%, producing gum with an average weight of 0.33 ± 0.08 g (mean ± SD, n = 34) during the same interval. The OSs demonstrated significantly greater gum secretion than the MSs (*p* = 0.0176).

### 2.3. Transcriptome Analysis of OSs, MSs, and HBs

To investigate the metabolic pathways involved in response to ALB gnawing and oviposition on Russian olive, we conducted transcriptome sequencing of ALBs’ OSs, MSs, and healthy branches (HBs) of Russian olive. The total number of clean reads obtained per sample through high-throughput sequencing varied from 40.82 million to 70.2 million. All samples had filtered Q30 values exceeding 93.00%, with a GC content ranging from 42.25 to 46.05%. The transcriptome data indicated that over 81.78% of the clean data mapped to the reference genome, with more than 75.61% of the reads mapping uniquely. These findings confirmed the reliability of the transcriptome data for identifying differentially expressed genes (DEGs).

#### 2.3.1. DEGs among OSs, MSs, and HBs

The principal component analysis (PCA) results revealed that principal components 1 and 2 accounted for 49.60% and 23.81% of the variance among treatment groups, respectively (Figure 3a). The MSs and OSs were distinguishable from the HBs at PC1, suggesting the presence of DEGs related to pests and mechanical injury responses. The FPKM statistical analysis indicated distinct gene expression patterns between OSs and MSs. Genes with FDR values <0.05 and |log2 (fold change)| > 1 were categorized as DEGs. Compared to the HBs, the OSs exhibited 8303 DEGs, comprising 4446 up-regulated and 3857 down-regulated genes. Similarly, the MSs presented 7664 DEGs, including 4311 up-regulated and 3353 down-regulated genes. Furthermore, the OSs displayed 5174 DEGs relative to the MSs, consisting of 2444 up-regulated and 2730 down-regulated genes (Figure 3b). Among these, 869 genes were commonly differentially expressed in the scars (Figure 3c), while 519 genes were uniquely differentially expressed in the OSs. Notably, the DEGs regulated by the OSs were the most numerous compared to those regulated by the HBs.

#### 2.3.2. KEGG Pathway Annotations of DEGs

To elucidate the impact of ALB oviposition on metabolic pathways compared to mechanical damage, we performed KEGG pathway analyses on DEGs between paired groups, revealing significant enrichment in OS processes. In the OS versus HB comparison, 19 of the top 20 KEGG pathways were metabolism-related, highlighting OSs’ substantial impact on metabolic processes (Figure 4a). In the MS versus HB comparison, the top 20 KEGG pathways included 17 metabolic pathways, 1 organismal systems pathway, and 2 environmental information processing pathways (Figure 4b), indicating that both damage types primarily activated pathways related to Russian olive metabolism. Relative to the MSs, the OSs exhibited significant enrichment in 12 KEGG pathways, 9 of which pertained to metabolism (Figure 4c). Following a *p*-value adjustment using the BH method, only two pathways retained significance: starch and sucrose metabolism (map00500) and photosynthesis (map00195). Thus, we infer that, unlike mechanical damage, ALB oviposition on Russian olive modifies photosynthesis and induces additional sugar metabolic responses.

### 2.4. Differences in Metabolic Components among OSs, MSs, and HBs

To investigate the metabolic pathways affected by ALB oviposition and mechanical damage, we conducted non-targeted metabolomic LC/MS sequencing of ALB OSs, MSs, and HBs on Russian olive.

#### 2.4.1. Differentially Accumulated Metabolites among OSs, MSs, and HBs

A total of 757 metabolites across 83 classes were identified, with carboxylic acids and derivatives, benzene and substituted derivatives, organooxygen compounds, fatty acyls, flavonoids, indoles and derivatives, and phenols being the most prevalent (Figure 5a). The PCA score plots of metabolites displayed distinct separation among the OS, MS, and HB groups, demonstrating high replication reliability. Principal components accounted for 27.9% (PC1) and 13.80% (PC2) (Figure 5b). Further analysis unveiled differentially accumulated metabolites (DAMs) among the OSs, MSs, and HBs (Figure 5c). Most DAMs were activated in response to OSs, suggesting their involvement in the resistance response elicited by ALBs.

#### 2.4.2. KEGG Pathway Annotations of DAMs

The KEGG analysis identified the top five pathways enriched in the OSs: biosynthesis of phenylpropanoids, biosynthesis of amino acids, biosynthesis of plant secondary metabolites, aminobenzoate degradation, and pyrimidine metabolism (Figure 6a). In the MSs, the most enriched pathways included biosynthesis of phenylpropanoids, biosynthesis of amino acids, flavonoid biosynthesis, biosynthesis of plant secondary metabolites, and protein digestion and absorption (Figure 6b). In contrast to the MSs, the DAMs in the OSs exhibited enrichment in the biosynthesis of phenylpropanoids, starch and sucrose metabolism, flavonoid biosynthesis, protein digestion and absorption, the cAMP signaling pathway, lysine degradation, morphine addiction, linoleic acid metabolism, GABAergic synapse, alcoholism, and beta-alanine metabolism (Figure 6c).

### 2.5. Differences in the Carbohydrates and Carbohydrate Conjugates among OSs, MSs, and HBs

The results revealed distinct sugar metabolism pathways between OSs and MSs. To further evaluate sugar differences in the gum of the Russian olive tree, we quantified the relative contents of 35 identified carbohydrates and carbohydrate conjugates related to carbohydrate metabolism (Figure 7). The major components of the tree gum include gluconic acid and D-mannose, which are more prevalent in OSs.

### 2.6. Differential Co-Expression of Genes and Metabolites of Russian Olive Induced by ALBs

Distinct metabolic responses were observed between the OSs and MSs. To assess the metabolic processes influenced by ALB activity, as opposed to just mechanical injury alone, we conducted a comprehensive analysis of the transcriptome and metabolome of both the OSs and MSs. A histogram illustrated the enrichment of KEGG pathways in DEGs and DAMs in the OSs compared to MSs (Figure 8). Among the DEGs and DAMs, 306 DEGs and 39 DAMs showed significant enrichment across 12 and 34 metabolic pathways, respectively. Notably, 116 DEGs and 25 DAMs were enriched in four major metabolic pathways: starch and sucrose metabolism, glutathione metabolism, flavonoid biosynthesis, and phenylpropanoid biosynthesis. Furthermore, starch and sucrose metabolism played a critical role in the response of Russian olives to OSs induced by ALBs.

### 2.7. Potential Candidate DEGs and DAMs in the Starch and Sucrose Metabolism Pathway

Fifty-two DEGs and seven differentially expressed metabolites (DEMs) in the starch and sucrose Metabolism pathway were analyzed comparing the OSs to MSs. We first visualized all DEGs in the starch and sucrose metabolism pathway (Figure 9a). A cluster analysis was performed on six replicate samples of OSs and MSs. In the OSs, 17 genes were significantly up-regulated, while 26 genes were down-regulated (Figure 9b). A further cluster analysis was conducted on all DEGs, resulting in the formation of 10 clusters, with the majority localized in clusters 2 and 3, as shown in Figure 9c,d. Cluster 2 consisted of 17 genes exhibiting the most significant up-regulation, many of which were co-localized, while Cluster 3 included 23 genes with the most pronounced down-regulation. Gene annotation revealed conserved structural motifs within the same gene families, including glycosyl hydrolase family 1, glycosyl hydrolases family 32 N-terminal domain, glycosyl transferases group 1, and alpha-amylase (Appendix A). The analysis suggested that ALB activity enhances the transcription of genes involved in starch and sucrose metabolism, potentially increasing the production of monosaccharides (e.g., D-glucose and D-fructose).

Additionally, seven differentially abundant metabolites (DAMs) were identified in the starch and sucrose metabolism pathway. The analysis indicated the significant up-regulation of uridine diphosphate glucose (UDPG), alpha-D-glucose-1-phosphate (α-D-glucose-1p), beta-D-glucose-1-phosphate (β-D-glucose-1p), cellobiose-6-phosphate (cellobiose-6′p), alpha-maltose 1-phosphate (α-maltose-1p), and glucose 6-phosphate (G6P), while the expression of sucrose was markedly reduced (Appendix A).

Among the seven DAMs identified in the OSs compared to the MSs, a pathway for the activation of sucrose hydrolysis, centered on α-D-glucose-1p and G6P, was elucidated (Figure 10a). Co-analysis with DEGs revealed the following: (1) sucrose in the OSs was degraded to UDPG under the regulation of *EaSUS1*; (2) the expression of *EaYfcE1*, responsible for converting UDPG to α-D-glucose-1p, was decreased, while overall levels of α-glucose-1p increased (presumably due to the conversion of a portion of G6P into α-glucose-1p mediated by *EaPGM1*). In the absence of DEGs, α-D-glucose-1p could convert into α-maltose-1p, β-D-glucose-1p, and cellobiose-6-phosphate, which were up-regulated and potentially converted into G6P. Upon analyzing the situation in the MSs versus HBs, we found no reduction in sucrose levels and no increase in UDPG levels (Figure 10b). However, up-regulation of *EaSUS2* and down-regulation of *EaSUS3* were observed, along with an increase in the expression of *EaUGP2*, which catalyzes the conversion of α-D-glucose-1p to UDPG. This was accompanied by a decrease in α-D-Glucose-1p, a decrease in β-D-glucose-1p, and an increase in G6P. The differences between OSs and HBs were not merely a composite of variations between OSs and MSs or between MSs and HBs (Figure 10c). Instead, the modulation of α-D-glucose-1p and β-D-glucose-1p maintained a balance: the down-regulation of sucrose and up-regulation of α-maltose-1p were not combined with significant reductions in cellobiose; the down-regulation of *EaSUS2* and *EaYfcE1*, along with the up-regulation of *EaPGM1*, were not enriched.

In summary, we identified a unique pathway specifically activated in ALB OSs, wherein three genes and four metabolites were implicated in a sucrose-activation process leading to UDPG, α-D-glucose-1p, and G6P (Figure 11). *EaSUS1* up-regulated sucrose to activate UDPG, while *EaYfcE1* down-regulated UDPG to α-D-glucose-1p. The interconversion between α-D-glucose-1p and G6P was facilitated by the translocase regulated by *EaPGM1*.

Transcriptome results were validated through qRT-PCR. We performed qRT-PCR analysis on candidate genes. As shown in Appendix A, the expression levels of six candidate genes varied across different types of scars, confirming the reliability of the transcriptome sequencing results.

### 2.8. Total Sugar, Total Polysaccharide, Sucrose, and Glucose Content of OSs, MSs, and HBs

The results indicated differences in carbohydrate composition between the OSs and MSs. To assess the precise changes in sugar content due to ALB infestation in Russian olive, we analyzed the concentrations of total sugar, total polysaccharide, sucrose, and glucose in the OS, MS, and HB samples (Figure 12). Significant increases in total sugar, total polysaccharide, and sucrose levels were observed in ALB-infested OSs compared to HBs [F(3, 8) = 178.5, *p* < 0.0001; F(3, 8) = 165.7, *p* < 0.0001; F(3, 8) = 133.7, *p* < 0.0001]. Furthermore, glucose contents in the OSs were comparable to those in the HBs, with both exceeding the levels found in the MSs. While total sugar content in the OSs was similar to that in the MSs, total polysaccharide content was higher in the OSs than in the MSs (Tukey’s multiple comparison test, *p* < 0.0001). Sucrose levels were greater in the MSs than in the OSs, whereas glucose levels were lower in the OSs.

## 3. Discussion

Russian olive attracts the ALB for oviposition and subsequently eliminates their eggs by secreting gum in the OSs [27]. Our findings demonstrated that scarring stimulates an increased production of Russian olive gum, and the tissue surrounding the wound exhibits elevated levels of sugar, polysaccharides, and sucrose. Notably, ALB oviposition inflicts distinct alterations in starch and sucrose metabolism that differ from those caused by mechanical damage.

Insects infesting plants trigger an insect resistance response that is distinct from the response to mechanical injury [28]. For instance, compared to mechanical damage, a significant enrichment of differential genes related to sugar metabolism was observed when *Populus* spp. are affected by *Saperda populnea* (Linnaeus, 1758). Research suggests that plants can physiologically detect and respond to the presence of herbivore insect eggs prior to hatching, representing a pre-ovipositional defense mechanism [29]. Following insect oviposition, plants possess the capacity to generate ovicidal compounds [30], which can lead to gall formation [31] or directly induce ovum necrosis [32]. We propose that the gum secreted by Russian olive in response to incisions made by ALBs represents an inducible insecticidal defense mechanism. This gum not only aids in sealing the wounds but also directly kills the eggs of ALBs.

Not all tree species have the ability to secrete gum; this trait is restricted to certain species, including *Hevea brasiliensis* (Willd. ex A. Juss.) Müll. Arg. [33], *Liquidambar formosana* Hance [34], and *Aquilaria sinensis* (Lour.) Spreng [35]. It is widely acknowledged that gums exhibit antimicrobial and wound-healing properties [36]. According to the results presented in Section 2.1 and Section 2.2, damage induces gum secretion in Russian olive, with higher gum levels observed in the OSs compared to the mechanical scars. This indicates that the insect-resistant response elicits a stronger gum secretion reaction. Thus, the abundance of Russian olive tree gum facilitates the elimination of ALB eggs.

Fluctuations in sugar concentration directly influence the osmotic potential of both cells and solutions. Changes in osmotic potential are critical for nutrient transport, which subsequently affect sugar movement, water potential, and water flow. Ultimately, these alterations impact a plant’s capacity to withstand unfavorable environmental conditions. For example, the starch and sugar content in tissues surrounding the wound site were altered following injury from *A. sinensis* [37]. This observation confirms a significant difference in sugar and starch amounts between the MS and OS samples compared to the HB sample in Section 2.8. As mentioned in the introduction, polysaccharides are the primary constituents of Russian olive tree gum. The total polysaccharide content was found to follow the order OSs > MSs > HBs, which aligns with our observations of gum production in Section 2.2. Therefore, the increase in polysaccharides in Russian olive tree gum may directly enhance gum production, possibly altering its viscosity and several hydrodynamic properties [38]. The source of this increased polysaccharide may be traced to findings from transcriptome and metabolome analyses within the starch and sucrose metabolic pathways. The observed reduction of sucrose in the OSs compared to the mechanical scars was also explained, as it is hydrolyzed to glucose (Figure 12). This process, which involves energy release, is intrinsically linked to gum formation.

The primary components of Russian olive gum include D-galactose, L-arabinose, L-rhamnose, gluconic acid, D-glucuronic acid, and D-mannose [18,38]. Our findings indicate that the hydrolysis of sucrose in Russian olive gum from OSs leads to the production of UDPG, α-D-glucose-1p, and G6P. UDPG serves as a vital intermediate in multiple metabolic pathways and biosynthetic activities, including the production of complex carbohydrates such as starch and glycogen, lipopolysaccharides, and glycosphingolipids [39,40,41]. G6P can be converted into glucose or can enter the glycolytic pathway for energy production and polysaccharide synthesis. Additionally, a comparative analysis of carbohydrate content in OSs revealed variations, including an increase in gluconic acid and D-mannose. Sucrose synthase, a key enzyme in sucrose metabolism, plays a significant role in regulating fruit quality and yield [42,43]. *EaSUS1* encodes a type of sucrose synthase. Phosphoglucomutase, crucial for starch synthesis, significantly influences carbon flux during triacylglycerol accumulation [44]. *EaPGM1* encodes a plastid isoform of phosphoglucomutase, which is essential for controlling carbon flux in photosynthesis [45]. A significant up-regulation of *EaSUS1* and *EaPGM1* was observed in the OSs, while *EaSUS1* was notably repressed in the MSs (Appendix A). *EaSUS1* is believed to play a pivotal role in modulating the gum response pattern, with its expression potentially influenced by ALB activity.

We present a theoretical model of the response mechanism in Russian olive to OSs in Figure 13, which is mediated by transcriptional and metabolic processes. The activation of ALB triggers the up-regulation of starch and sucrose metabolism genes in Russian olive. Key genes, such as *EaSUS1* and *EaPGM1*, enhance the production of UDPG, α-D-glucose-1p, and G6P, potentially playing a crucial role in the plant’s response to OSs by modulating polysaccharide production. These findings lay the groundwork for understanding the response mechanism of Russian olive to OSs. However, further investigation is necessary to fully elucidate the operations and biochemical processes of alternative pathways, as well as the specific genes and metabolites involved in this route.

Figure 13 shows the proposed model for the response of Russian olive trees to OSs caused by ALBs. In damage-associated molecular patterns (DAMPs), sucrose is unloaded from the phloem. (1) Sucrose is hydrolyzed by invertase (INV) into glucose, which is then converted by hexokinase (HXK) to G6P. (2) Sucrose, UDPG, and G1P are mutually transformed by sucrose synthase (SUS), phosphodiesterase (YfcE), and small ribosomal subunit protein (RT02). In HAMPs, additional sucrose is degraded by SUS to UDPG, while the conversion of UDPG to G1P is inhibited by YfcE. More G6P is derived from G1P through PGM.

Additionally, “dead-end trapping technology” is an ecological control strategy that employs dead-end trap trees to attract pests and efficiently eliminate their offspring. This approach relies on two critical elements: effective attraction and efficient pest removal. Firstly, an effective dead-end trap tree must exhibit a strong lure for adult pests, qualifying as a trap tree based primarily on its attraction efficiency, even in the absence of inherent killing capabilities. Secondly, it must also effectively eliminate pest progeny. A tree lacking sufficient or potent attractant properties is merely classified as highly resistant. Furthermore, an exemplary dead-end trap tree should possess additional attributes such as robust resilience, ease of cultivation, cost effectiveness, and lasting efficacy, factors that are essential for the success of sustainable forestry practices.

## 4. Materials and Methods

### 4.1. Sampling Site and Sample Preparation

The sampling site was situated in a plantation (39.76° N, 98.25° E) along Jiadun Road near the “First Pier of the Great Wall” in Jiayuguan City, Gansu Province. The sample plot included seven rows of *Populus euphratica*, five rows of Russian olive, and one row of *Salix laevigata*. The area was transplanted in 2014, irrigated, and regularly maintained, but was susceptible to damage from the ALB. The Russian olive trees are 12–15 years old, approximately 4.2 m in height, and have a diameter at breast height of about 11.20 cm. Ten replicates of weighed blocks of external gum were collected from carvings made in previous years.

The collection method for the OSs and MSs involved several steps: (1) surveying 10 m × 10 m sample plots to locate and mark all existing OSs on Russian olive trees; (2) marking any newly generated OSs within 24 h and creating MSs around the OSs using carving knives and tweezers to replicate the behavior of ALB females during oviposition; (3) after one day, collecting bark from the cambial region of the OSs and MSs using sterilized carving knives, along with HBs from the bark to the current year’s wood surface. Each OS sample was inspected for the presence of eggs, which were removed during collection. The collected samples were immediately placed in dry ice and transported under a cold chain to the laboratory at Beijing Forestry University for parameter analysis. Fifty days post-treatment, gum secretion cessation was confirmed prior to calculating the gum secretion rate, defined as (Gum secretion rate = number of gum secretion scars/total number of scars × 100%). Gum blocks were collected and weighed(Quintix224-1CN,), with ten replicates for each group.

### 4.2. High-Throughput Transcriptomics Analysis

The OSs, MSs, and HBs collected in Section 4.1 were utilized for transcriptomic profiling analysis. The Russian olive genome was obtained from Shen Xiang et al. (2020) [9]. The RNA-seq transcriptome library for Russian olive was constructed using 1 μg of total RNA in accordance with the Illumina^®^ Stranded mRNA Prep, Ligation kit (Illumina, San Diego, CA, USA). To identify differentially expressed genes (DEGs) between two samples, transcript expression levels were calculated using the transcripts per million reads (TPM) method. RSEM [4] was employed for quantifying gene abundances. Differential expression analysis was conducted using DESeq2 or DEGseq. DEGs exhibiting |log2FC| ≥ 1 and FDR ≤ 0.05 (DESeq2) [46] or FDR ≤ 0.001 (DEGseq) [47] were deemed significantly expressed. Additionally, functional enrichment analysis, the Kyoto Encyclopedia of Genes and Genomes (KEGG), was performed to identify significantly enriched DEGs in metabolic pathways, with a Bonferroni-corrected *p*-value ≤ 0.05 compared to the whole transcriptome background. The KEGG pathway analysis utilized KOBAS (Version 3.0, Beijing, China) [48]. RNA purification, reverse transcription, library construction, and sequencing were carried out at Shanghai Majorbio Bio-pharm Biotechnology Co., Ltd. (Shanghai, China) following the manufacturer’s instructions (Illumina, San Diego, CA, USA). The concentration and insert size of the library were assessed using Qubit 2.0 (Life Technologies, Carlsbad, CA, USA) and Agilent 2100 (Agilent Technologies, Santa Clara, CA, USA), respectively. The effective concentration of the library was quantified using qPCR.

### 4.3. Metabolomics Profiling Analysis

Untargeted metabolomics profiling of the OSs, MSs, and HBs was executed using LC/MS techniques. The methodology involved precise measurement of the sample, which was placed in a 2 mL centrifuge tube. A volume of 600 µL of pre-cooled (−20 °C) methanol (MeOH) containing 4 ppm of -Amino-3-(2-chloro-phenyl)-propionic acid was added. The mixture was vortexed for 30 s (BE2600, Haimen Kylin-Bell Lab Instruments Co., Ltd., Haimen, China), followed by the addition of 100 mg of glass beads and processing in a tissue grinder for 90 s at 60 Hz(MB-96, MEIBI, Zhejiang, China). Subsequently, the sample underwent ultrasound treatment at room temperature for 15 min, followed by centrifugation at 12,000 rpm at 4 °C for 10 min(H1850-R, Cence, Changsha, China). The supernatant was filtered using a 0.22 μm membrane and transferred to a detection vial for subsequent LC-MS analysis.

A chromatographic analysis was performed using a Vanquish UHPLC System (Thermo Fisher Scientific, Waltham, MA, USA) equipped with an ACQUITY UPLC^®^ HSS T3 column (150 mm× 2.1 mm, 1.8 µm) (Waters, Milford, MA, USA) maintained at 40 °C. The flow rate and injection volume were standardized at 0.25 mL/min and 2 μL, respectively. For the LC-ESI (+)-MS analysis, the mobile phases were composed of 0.1% formic acid in both acetonitrile and water. Conversely, the LC-ESI (−)-MS analysis utilized acetonitrile and 5 mM ammonium formate. Metabolite detection was performed using a Q Exactive HF-X mass spectrometer (Thermo Fisher Scientific, USA) equipped with an ESI ion source, employing simultaneous MS1 and MS/MS acquisition (Full MS-ddMS2 mode, data-dependent MS/MS). These experimental procedures were conducted at Suzhou Panomics Biomedical Technology Co., Ltd. (Suzhou, China).

### 4.4. Correlation Network Analysis of Transcriptomics and Metabolomics Data

A correlation network analysis was performed to examine the relationships between metabolites and genes, particularly within shared KEGG pathways. Correlation coefficients were calculated based on gene expression and metabolite abundance, and gene–metabolite pairs were ranked by their absolute values (WPS Office, version 2024v17827, Kingsoft Office Software Co., Ltd., Beijing, China).

### 4.5. Total Sugar, Total Polysaccharide, Sucrose, and Glucose Content

The DNS colorimetric method was utilized to quantify the total sugar content in the samples, wherein acid hydrolysis was employed to convert total sugars into reducing sugars. Co-heating with the DNS reagent under alkaline conditions resulted in the formation of orange-red-colored amino compounds in an excess NaOH solution, with a maximal absorption peak at 540 nm (Synergy H1M, Sartorius AG, Göttingen, Germany).

Total polysaccharide content was extracted using the aqueous alcoholic precipitation method and quantified using the phenol-sulfuric acid method, which exhibited a maximum absorption peak at 490 nm.

Sucrose quantification was accomplished using the resorcinol method, where the initial decomposition of reducing sugars in the sample occurred through co-heating with an alkali. Subsequent hydrolysis of sucrose under acidic conditions yielded glucose and fructose. The fructose then reacted with resorcinol to produce a colored substance with a characteristic absorption peak at 480 nm.

Targeted oxidation of glucose generated a red product, which reacted with a chromogenic agent, displaying a maximal absorption peak at 520 nm, thus allowing for the determination of glucose content.

### 4.6. Statistical Analysis

Data from Section 4.1 and Section 4.5 were analyzed and visualized using Graphpad Prism 9.5.0 (Boston, MA, USA, GraphPad Software) employing the Mann–Whitney test, Kruskal–Wallis test, Dunn’s test, Tukey’s multiple comparison test, and one-way ANOVA. The high-throughput transcriptomics data were analyzed and visualized using the Majorbio Cloud Platform (www.majorbio.com), accessed between 16 September 2022 and 4 July 2024. Identified metabolites and genes underwent metabolic pathway analysis utilizing MetaboAnalyst 4.0 (http://www.metaboanalyst.ca), accessed between 26 December 2022 and 1 November 2023. Differentially abundant metabolites (DAMs) were defined as exhibiting a log2 fold change (FC) ≥ 2 or FC ≤ 0.5, alongside variable importance in projection (VIP) scores > 1. The identified metabolites in the metabolomics were subsequently mapped to the KEGG pathway (http://www.genome.jp/kegg/ )accessed on 9 December 2022. [11]. Principal component analysis (PCA) was employed to identify significantly different metabolite levels (*p*-value < 0.05). Data analysis and graphical representations were conducted using WPS Office (version 2024v17827, Kingsoft Office Software Co., Ltd., Beijing, China).

## 5. Conclusions

This study presents the first analysis integrating the metabolome and transcriptome of Russian olive trees damaged by the Asian long-horned beetle (ALB). We identified a total of 8303 differentially expressed genes (DEGs) through transcriptome sequencing and detected 757 differentially accumulated metabolites (DAMs) in the oviposition scars (OSs), mechanical scars (MSs), and healthy branches (HBs) of Russian olive. A network interaction analysis revealed four significant metabolites (sucrose, UDPG, G1P, and G6P) and three key hub genes (*EaSUS1*, *EaYfcE1*, and *EaPGM1*) involved in the starch and sucrose metabolism pathway associated with OSs. These substances and genes are proposed to model the response mechanism of Russian olive to ALB oviposition scars. This study lays a critical foundation for future research on the molecular mechanisms underlying the attraction and eradication of ALB by Russian olive trees.

## Figures and Tables

**Figure 1 ijms-25-09504-f001:**
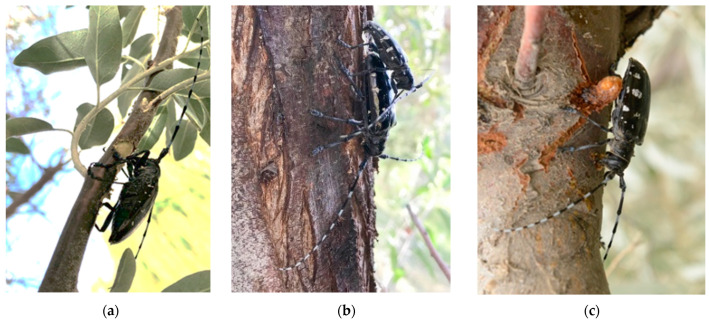
The oviposition process of Asian long-horned beetles on Russian olives: (**a**) an adult beetle supplements nutrients on a twig; (**b**) adult beetles mate on a branch; (**c**) the female beetle uses her mouthparts to chew an oviposition scar; (**d**) subsequently, the female beetle rotates and uses an ovipositor to penetrate the scar for oviposition; (**e**) inside the scar, an egg encased in the transparent gum; (**f**) a comprehensive image of Russian olive.

**Figure 2 ijms-25-09504-f002:**
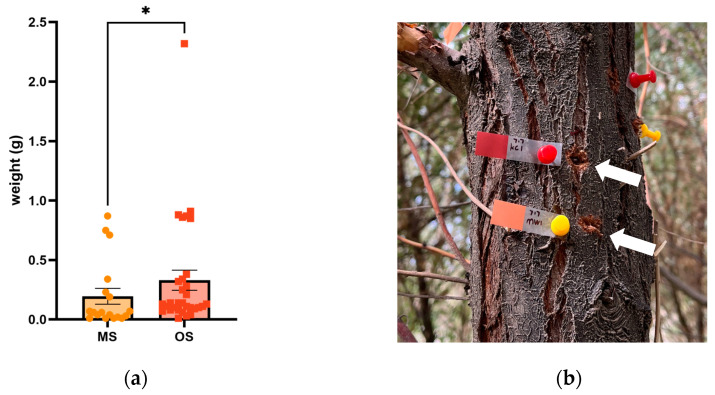
Gum secreted from MSs and OSs: (**a**) weight of gum from MSs and OSs; (**b**) field images of the two adhesive treatments. Note: * indicates statistically significant differences (*p* < 0.05) as determined by the Mann–Whitney test. The red label denotes OSs, and the orange label denotes MSs.

**Figure 3 ijms-25-09504-f003:**
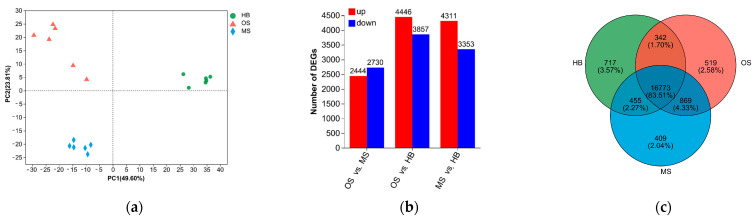
DEGs in the transcriptomes of ALBs’ OSs, MSs, and HBs on Russian olive: (**a**) PCA plots; (**b**) illustration of significantly up- and down-regulated DEGs; (**c**) Venn diagram of common and specific DEGs.

**Figure 4 ijms-25-09504-f004:**
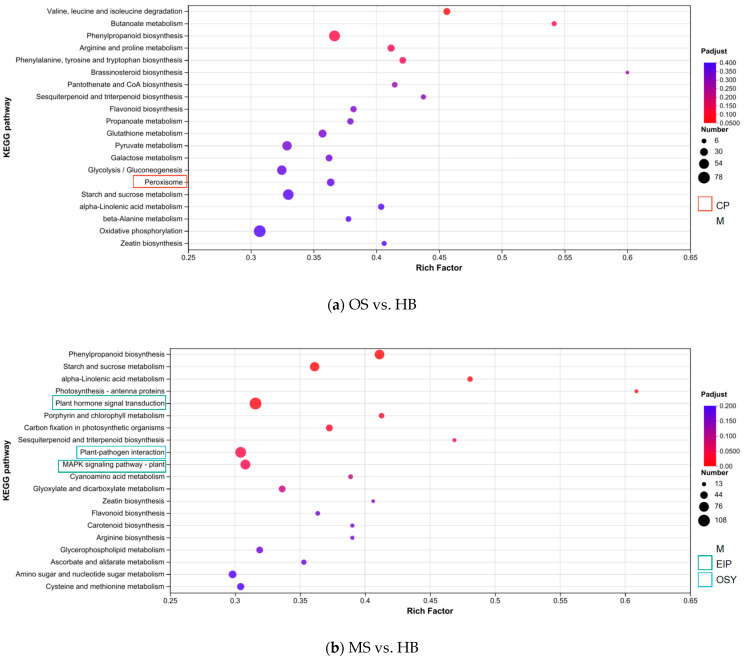
Top 20 KEGG pathways in enrichment analysis of DEGs in OSs, MSs, and HBs of Russian olive: (**a**) OSs vs. HBs; (**b**) MSs vs. HBs; (**c**) OSs vs. MSs. The top 20 KEGG pathways are listed based on the Rich factor [the ratio of enriched genes in the pathway (sample number) to annotated genes (background number)], with the dot size representing the gene count and color indicating the *p*-value range. The branches of the KEGG metabolic pathways names are represented by different colors, which include metabolism (M), genetic information processing (GIP), environmental information processing (EIP), cellular processes (CPs), organismal systems (OSYs), human diseases (HDs), and drug development (DD).

**Figure 5 ijms-25-09504-f005:**
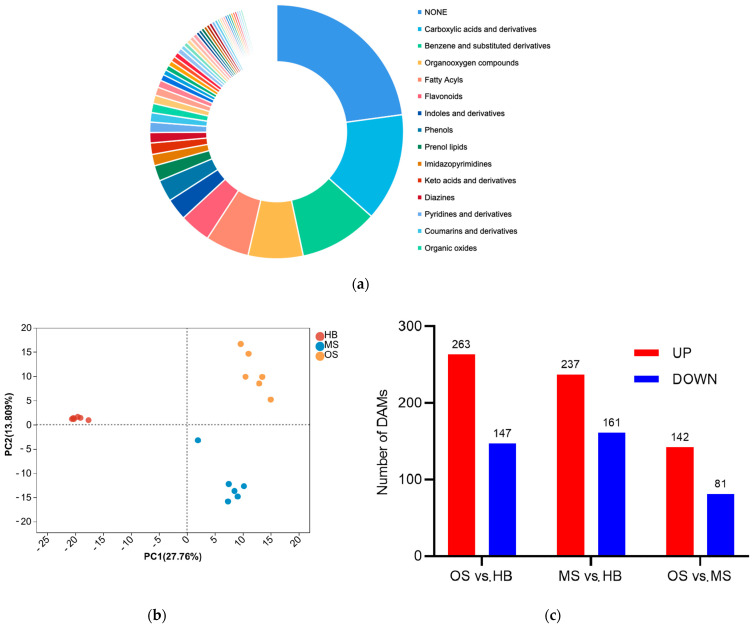
Qualitative and quantitative analyses of the metabolome of OSs, MSs, and HBs: (**a**) number of metabolites in each category; (**b**) PCA of OSs, MSs, and HBs: (**c**) number of DAMs (|FC| ≥ 2) in OSs vs. HBs, MSs vs. HBs, OSs vs. MSs.

**Figure 6 ijms-25-09504-f006:**
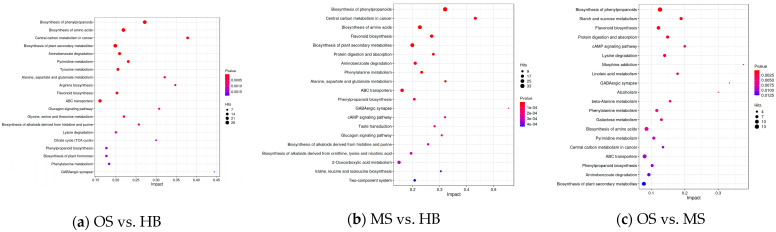
Top 20 lists of KEGG pathways in DAMs: (**a**) OSs vs. HBs; (**b**) MSs vs. HBs; (**c**) OSs vs. MSs. Note: The vertical axis represents the pathway name; the horizontal axis represents the Rich factor, the ratio of enriched genes in the pathway to annotated genes; and a higher Rich factor indicates greater enrichment. The dot size denotes gene count, and the dot color signifies *p*-value ranges.

**Figure 7 ijms-25-09504-f007:**
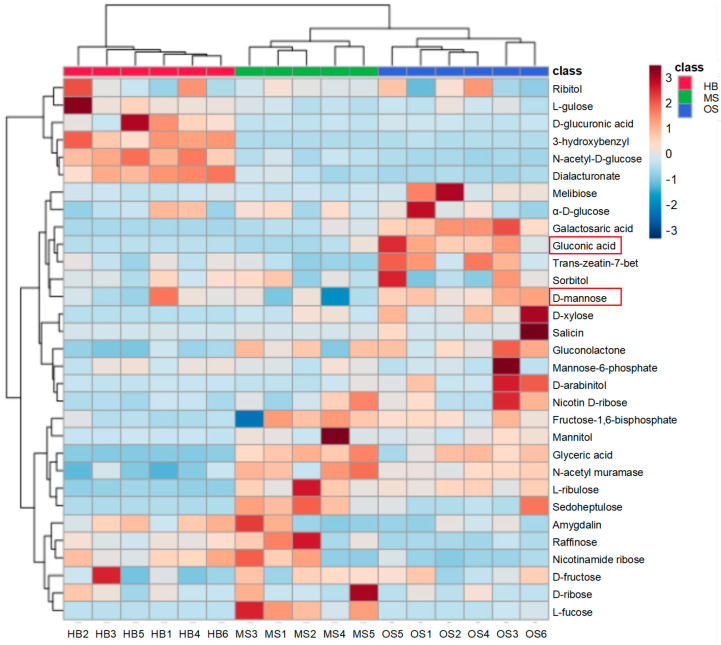
The heatmap illustrates the saccharide content in OSs, MSs, and HBs. Different colors indicate the relative accumulation of DAMs, normalized using log2 fold change (log2FC). Red indicates the presence of the same major components of tree gum.

**Figure 8 ijms-25-09504-f008:**
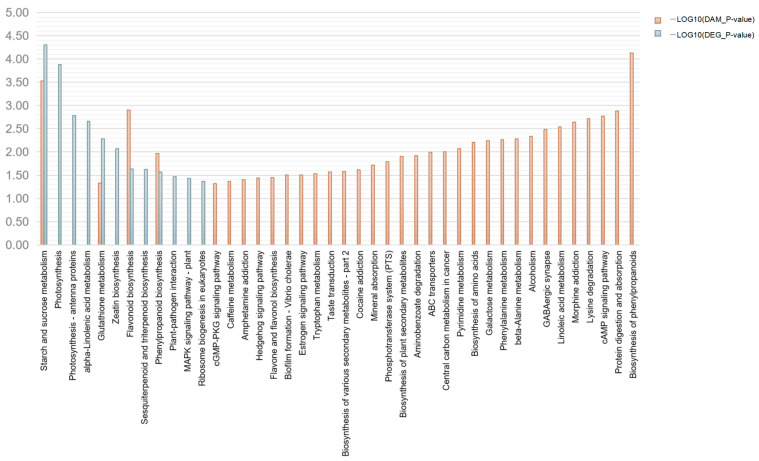
KEGG pathway enrichment analysis of DEGs and DAMs in the transcriptome and metabolome. Note: the horizontal axis represents metabolic pathways; the vertical axis represents the enriched *p*-values of DEGs (blue) and DAMs (orange) using −logP.

**Figure 9 ijms-25-09504-f009:**
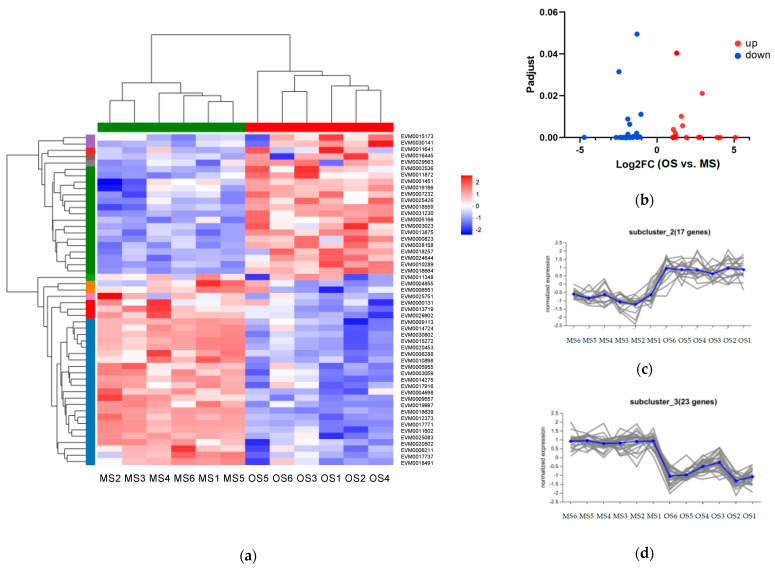
Gene expression patterns of DEGs between OSs and MSs: (**a**) heatmap of significant DEGs; (**b**) volcano plot of significant DEGs. Note: Blue indicates down-regulation and red indicates up-regulation. The vertical coordinate represents Padjust (significance level); the heatmap of gene expression is normalized in log2FC, and the clusters are defined based on Euclidean distance. (**c**,**d**) Gene expression analysis based on expression levels. Note: the number of genes in each cluster is indicated in brackets. Each line in the figure represents the change trend of a gene, and the blue line represents the change trend of the average expression level of all genes in the gene set.

**Figure 10 ijms-25-09504-f010:**
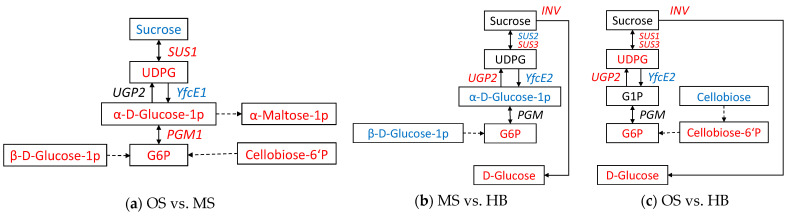
Differentially enriched metabolites and genes in a glycolytic pathway: (**a**) OS vs. MS; (**b**) MS vs. HB; (**c**) OS vs. HB. Note: Boxes represent metabolites, solid lines and letters represent differentially enriched genes. Red indicates up-regulation and blue indicates down-regulation. Dashed lines and letters indicate that the metabolite or gene was not differentially enriched.

**Figure 11 ijms-25-09504-f011:**
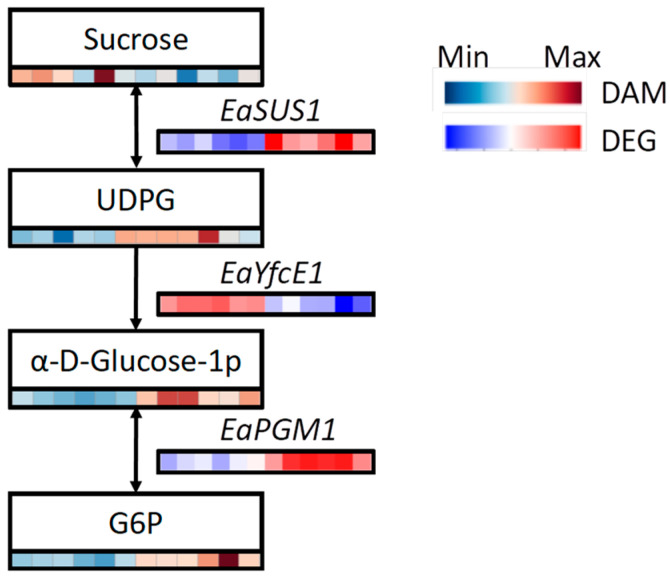
Integration analysis of DEGs and DAMs linked to the starch and sucrose metabolism pathway. The relative expression levels of DEGs and the accumulation of DAMs were calculated using log2FC.

**Figure 12 ijms-25-09504-f012:**
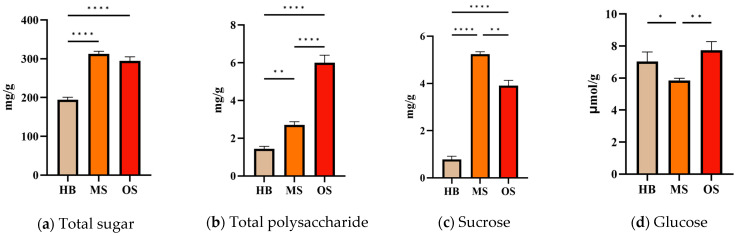
Total sugar, total polysaccharide, sucrose, and glucose content in OSs, MSs, and HBs: (**a**) total sugar content; (**b**) total polysaccharide content; (**c**) sucrose content; (**d**) glucose content. Note: asterisks denote significant differences from control HBs according to Tukey’s multiple comparison test (n = 3, * *p* < 0.05, ** *p* < 0.01, **** *p* < 0.0001).

**Figure 13 ijms-25-09504-f013:**
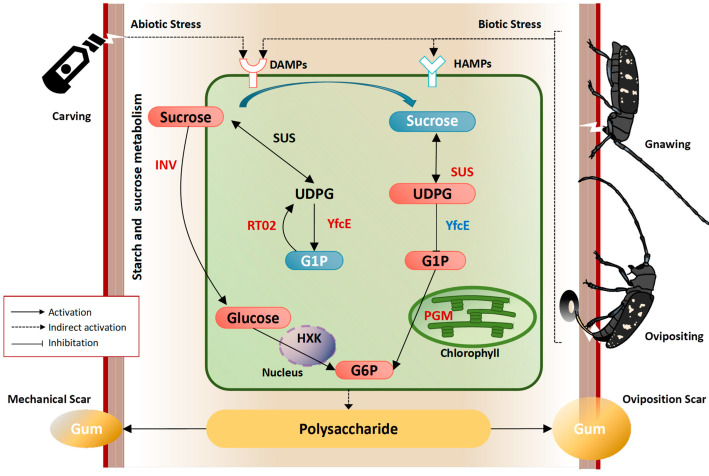
Proposed model for the response of Russian olive trees to oviposition scar caused by Asian long-horned beetles. In damage-associated molecular patterns (DAMPs), sucrose is unloaded from the phloem. (1) Sucrose is hydrolyzed by invertase (INV) into glucose, which is then converted by hexokinase (HXK) to glucose-6-phosphate (G6P). (2) Sucrose, uridine diphosphate glucose (UDPG), and D-glucose-1-phosphate (G1P) are mutually transformed by sucrose synthase (SUS), phosphodiesterase (YfcE), and small ribosomal subunit protein (RT02). In herbivore-associated molecular patterns (HAMPs), additional sucrose is degraded by SUS to UDPG, while the conversion of UDPG to G1P is inhibited by YfcE. More G6P is derived from G1P through phosphoglucomutase (PGM).

## Data Availability

The data presented in this study are available on request from the corresponding author.

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
