# Peer review of "Interactions at the Oviposition Scar: Molecular and Metabolic Insights into *Elaeagnus angustifolia*’s Resistance Response to *Anoplophora glabripennis"

_ijms, 2024, doi:10.3390/ijms25179504_

Round 1
Reviewer 1 Report
Comments and Suggestions for Authors
1. The manuscript “Interactions at the oviposition scar: molecular and metabolic insights into Elaeagnus angustifolia's resistance response to Anoplophora glabripennisis” is a well conducted experimental study supported by a wealth of data. The authors performed several experiments to understand how the Russian olive gum secretions have an impact on the oviposition of Asian longhorned beetle and what the differences are in terms of gene expression and metabolites of gum secreted during oviposition scar, mechanical scar and healthy branch.
2. The authors have provided relevant literature on the background of the study however the abstract and the introduction lack the broader significance/implication of the study. Please consider adding it to the manuscript.
3. Are there any examples of dead-end trap trees currently used in forestry? If so, please provide some information on those examples.
4. The manuscript contains a high number of figures. It is advisable to keep the number under 10 and the additonal figures can go to supplementary figures. For example, Figure 1 and 2 can be moved to supplemental information file. Additionally, there is an inconsistency in the figure number in the legend and the text. Please maintain accuracy in the text particularly in lines 334, 387, 410 where the figure numbers need to be corrected.
5. Please mention in the legend of figure 2 the significance of the plus and minus sign on the arrows. If red and blue lines are sufficient to explain the purpose, these signs can be removed.
6. The information on plant defenses in lines 95 to 106 is very well explained however, this appears as general information, please ensure to relate the plant defense mechanisms in context of the current study.
7. Please mention in Figure 10a, if the heatmap of gene expression corresponds to normalized Log2FC? How does the clustering in the heatmaps done? What does it infer? A clear explanation of these points will enhance the readers' understanding.".
8. Please elaborate on how the information on DEGs in lines 151-160 is helpful and what it reveals. Providing more context on the significance and implications of the DEGs will strengthen this section.
Author Response
Thank you very much for taking the time to review our manuscript. We have made changes based on your suggestions. Please find the detailed responses below and also in track changes in the re-submitted files.
|
Comments 1: The manuscript “Interactions at the oviposition scar: molecular and metabolic insights into Elaeagnus angustifolia's resistance response to Anoplophora glabripennisis” is a well conducted experimental study supported by a wealth of data. The authors performed several experiments to understand how the Russian olive gum secretions have an impact on the oviposition of Asian longhorned beetle and what the differences are in terms of gene expression and metabolites of gum secreted during oviposition scar, mechanical scar and healthy branch. |
|||
|
Response 1: Thank you for your thorough review and for recognizing the value of our study. We appreciate your acknowledgment of our efforts in performing and presenting these experiments. We are grateful for your constructive feedback and would like to address the points you’ve raised.
|
|||
|
Comments 2: The authors have provided relevant literature on the background of the study however the abstract and the introduction lack the broader significance/implication of the study. Please consider adding it to the manuscript. |
|||
|
Response 2: Thank you for your suggestion. We have added “This study investigates the initial use of dead-end trap trees to manage invasive borer pests in forestry. The findings could provide important insights into the ecological self-control mechanisms of Asian long-horned beetle outbreaks.” “Our study focuses on the first pair of dead-end trap trees and invasive borer pests in forestry, potentially offering valuable insights into the ecological self-regulation of Asian long-horned beetle outbreaks.” (L26) to the Abstract and “This is the first documented case of a dead-end trap tree being used to manage invasive borer pests in forestry..” (L80) to the Introduction, hoping that they will adequately complement the broader importance of our study. |
|||
|
Comments 3: Are there any examples of dead-end trap trees currently used in forestry? If so, please provide some information on those examples. |
|||
|
Response 3: Thank you for pointing this out. Russian olive and ALB are the first pair of dead-end trap trees and pests in forestry, this is also the key innovation point of our studies. However, the dead-end plant are researched and reported in Agriculture, we reviewed them at 71-79 section. |
|||
|
Comments 4: The manuscript contains a high number of figures. It is advisable to keep the number under 10 and the additonal figures can go to supplementary figures. For example, Figure 1 and 2 can be moved to supplemental information file. Additionally, there is an inconsistency in the figure number in the legend and the text. Please maintain accuracy in the text particularly in lines 334, 387, 410 where the figure numbers need to be corrected. |
|||
|
Response 4: Thank you for your suggestions. We appreciaed for the corrections. We have moved the original Figure 2 in the attached Figure S1. We also checked and corrected the figure numbers throughout the whole manuscript, including lines 334, 387 and 410. As we undersatnd, Figure 1 is crucial to this article because it visually represents this biological phenomenon for the first time, aiding readers in better understanding the entire biological process. Additionally, another reviewer suggested that we include an image of the whole Elaeagnus angustifolia plant, so we hope to retain this figure to further enhance the comprehension of the biological phenomenon described in the paper. Thank you for your understanding.
|
|||
|
Comments 5: Please mention in the legend of figure 2 the significance of the plus and minus sign on the arrows. If red and blue lines are sufficient to explain the purpose, these signs can be removed. |
|||
|
Response 5: Thanks for your suggestion, we have removed the plus and minus signs, it is Figure. S1 in the revised version. |
|||
|
Comments 6: The information on plant defenses in lines 95 to 106 is very well explained however, this appears as general information, please ensure to relate the plant defense mechanisms in context of the current study. |
|||
|
Response 6: Thanks for your suggestion, we related the plant defense mechanisms with our reaserch as following, “Thus, we hypothesize that Russian olive may respond differently to OS compared to MS, with gum secretion potentially regulated by defense mechanisms associated with HAMPs.” at L108-110. |
|||
|
Comments 7: Please mention in Figure 10a, if the heatmap of gene expression corresponds to normalized Log2FC? How does the clustering in the heatmaps done? What does it infer? A clear explanation of these points will enhance the readers' understanding." |
|||
|
Response 7: Thanks for your suggestion. The heatmap of gene expression corresponds to the standardized Log2FC and the clusters in the heat map are analyzied based on the Euclidean distance, we added these information in L278-279. Its function is to classify the up- and down-regulation trends, so that readers can clearly see the trends of each gene, hoping to enhance readers' understanding. |
|||
|
Comments 8: Please elaborate on how the information on DEGs in lines 151-160 is helpful and what it reveals. Providing more context on the significance and implications of the DEGs will strengthen this section. |
|||
|
Response 8: Thanks for your suggestion, We provide more information on the significance and impact of DEGs in L154-155 and L163-164, as”These DEGs indicated that there were distinct gene expression patterns between OS and MS.””Among these, the DEGs regulated by OS are the most numerous in comparison to those regulated by HB.” at the same time, detailed description and biological analysis of this content are on line 169-183. |
|||
Reviewer 2 Report
Comments and Suggestions for Authors
Dear authors
Happy day
The paper is fine but need some improvement
General points
1- Kindly if you abbreviate some words use the abbreviation in the entire text. Fill name should be used just one time.
2- Kindly add the name of the software of all the software you have used.
3- Kindly use arrows to point the part of the image you interested to attract the attention to it.
4- Kindly add a new sub-title for the statistical analysis and include the test(s) you have used, software, etc.
5- Show the full image of the olive oil tree and why the insect select the hard woody part to infect it?
6- Describe the name of the crops which the Anoplophora glabripennis.
In the abstract part
7- Kindly avoid using abbreviation in the abstract or use in addition the full name.
In the introduction part
8- In figure 1 kindly use the same numbering style in the above text or vise versa.
9- If I well understand in figure (e) the egg(s) is trapped by the gyp, you have removed part of the tree surface to show the hidden eggs; and perhaps the gup is transparent; kindly clarify that all to the leader by rephrasing the legend of Figure 1.
10- In figure 2 kindly use “-“ to link uncomplete words in different liens.
In the results part
11- Kindly use arrows to spot what you need from the reader to see. You might magnify the important par.
12- Kindly add the name of the used software used for the results analysis.
13- Kindly include each finding represented in each image. In other words link the many images you have used in the text.
In the discussion part
14- (Optional) Kindly show that the tree did not made the same response if injured by an inert material.
In the material and method part
15- Describe the name of the used software.
16- Add a separate part to the statistical analysis.
In the references part
17- (Optional) Kindly add new and more references
In general: Your great observation which is highlight interested to the many should not let you miss some essential points in paper writing like the full description of the methods you used and the name of the software.
with my pleasure
Author Response
|
Thank you very much for taking the time to review our manuscript. We have made changes based on your suggestions. We apologize for the oversight in the description of the methods and software used. We have made the necessary revisions based on your suggestions. Please find the detailed responses below, as well as in the track changes of the resubmitted files.
|
||
|
Comments 1: Kindly if you abbreviate some words use the abbreviation in the entire text. Fill name should be used just one time. |
||
|
Response 1: Thank you for pointing this out. We appreciated for this comment. Therefore, we have thoroughly reviewed and . We have revised and standardized the abbreviations throughout the manuscript, keep the full name , when they appear for the first time |
||
|
Comments 2: Kindly add the name of the software of all the software you have used. |
||
|
Response 2: Thank you for pointing this out. We have added all the name and versions of the software we had used. |
||
|
Comments 3: Kindly use arrows to point the part of the image you interested to attract the attention to it. |
||
|
Response 3: Thank you for pointing this out. We have added arrows to point the secretion in this Figure at L132 to 135 .
|
||
|
Comments 4: Kindly add a new sub-title for the statistical analysis and include the test(s) you have used, software, etc |
||
|
Response 4: Thanks for your suggestion, We have followed your suggestion to add data analysis methods and software, which were previously dispersed in 4.1-4.5 according to the corresponding content, and have integrated the relevant content into the new subsection 4.6, “Data Analysis”. 4.6 Statistical analysis The data of 4.1 and 4.5 were annalyzed and visualized using Graphpad Prism 9.5.0 (The United States,Boston, GraphPad Software) with Kruskal-Wallis test, Dunn's test and One-way ANOVA. The data of high-throughput transcriptomics were analyzed and visualized using the Majorbio Cloud Platform online platform (www.majorbio.com). The identified metabolites and genes were subjected to metabolic pathway analysis utilizing MetaboAnalyst 4.0 (http://www.metaboanalyst.ca) for further investigation. DAMs were designated as a log2 fold change (FC) ≥ 2 and FC ≤ 0.5, along with variable importance in projection (VIP) scores > 1. Differential metabolites were subjected to pathway analysis The identified metabolites in metabolomics were then mapped to the KEGG pathway (http://www.genome.jp/kegg/)[11]. The metabolites and corresponding pathways were visualized using KEGG Mapper tool. Principal component analysis (PCA) were utilized to identify significantly different metabolite levels (p-value < 0.05). |
||
|
Comments 5: Show the full image of the olive oil tree and why the insect select the hard woody part to infect it? |
||
|
Response 5: Thanks for your suggestion, we showed the full image of the Russian olive tree in Figure 1f as followed. The reason of the insect select the hard woody part to infect it is the oviposition of adult ALB in the cambium of large branches facilitates the burrowing of its larvae into the xylem of the tree trunk, in this case there would be enough space niche for the larvae to develop We added these informations in L37: “ALB inflicts damage on various broad-leaved trees, including Populus, Salix, Betula, Acer, Ulmus, primarily through larval feeding in branch. After feeding and mating, female ALB choose to lay their eggs on the trunk and larger branches.” |
||
|
Comments 6 Describe the name of the crops which the Anoplophora glabripennis. |
||
|
Response 6 Thanks for your suggestion, we have revised and added the information as “ALB inflicts damage on various broad-leaved trees, including Populus, Salix, Betula, Acer, Ulmus (L37).” |
||
|
Comments 7 Kindly avoid using abbreviation in the abstract or use in addition the full name. |
||
|
Response Thanks for your suggestion, we used the full name in the abstract. The Russian olive (Elaeagnus angustifolia), functioning as a "dead-end trap tree" for Asian long-horned beetle (ALB), (Anoplophora glabripennis) in mixed plantations, can successfully attract the Asian long-horned beetle for oviposition and subsequently kill the eggs by gum. This study aimed to investigate gum secretion differences by comparing molecular and metabolic across three conditions: the oviposition scar, mechanical scar, and healthy branch using high performance liquid chromatography and high-throughput RNA sequencing methods. Our findings indicated that the gum mass secreted by the oviposition scar was 1.65 times greater than mechanical scar. Significant differences in gene expression and metabolism were observed among the three comparison groups. Kyoto encyclopedia of genes and genomes annotation and enrichment analysis showed that oviposition scar significantly affects starch and sucrose metabolism, leading to the discovery of 52 differentially expressed genes and 7 differentially accumulated metabolites. Network interaction analysis of differentially expressed metabolites and genes showed that EaSUS1, EaYfcE1, and EaPGM1 regulate sucrose, Uridine Diphosphate Glucose, α-D-Glucose-1P, and D-Glucose-6P. Through quantified the polysaccharide content in the OS was 2.22 times higher than MS, but the sucrose was less. The results indicated that, Asian long-horned beetle caused Russian olive sucrose degradation and D-Glucose-6P formation. Therefore, we hypothesize that, Asian long-horned beetle damages could enhance tree gum secretion through hydrolyzed sucrose, and stimulate the Russian olive's specific immune response. Our study focuses on the first pair of dead-end trap trees and invasive borer pests in forestry, potentially offering valuable insights into the ecological self-regulation of Asian long-horned beetle outbreaks.
|
||
|
Comments 8 In figure 1 kindly use the same numbering style in the above text or vise versa. |
||
|
Response 8 Thanks for your suggestion. We have revised and used the same numbering style in Figure 1(L64)
|
||
|
Comments 9If I well understand in figure (e) the egg(s) is trapped by the gyp, you have removed part of the tree surface to show the hidden eggs; and perhaps the gup is transparent; kindly clarify that all to the leader by rephrasing the legend of Figure 1. |
||
|
Response 9 Thank you for your insightful comments. We have removed the bark, and indeed, the gum is transparent. We have revised the figure legend as per your suggestion (L54 “(e) An ALB egg pit on Russian olive without the bark, and an egg is wrapped by transparent gum.”). We appreciate your suggestion.
|
||
|
Comments 10 In figure 2 kindly use “-“ to link uncomplete words in different liens. |
||
|
Response 10 Thank you for your suggestion. We have corrected the incomplete words and linked the terms across different lines using a hyphen. In response to the recommendations from other reviewers, this information is now provided as a supplementary figure (see FS1) or presented below:
|
||
|
Comments 11- Kindly use arrows to spot what you need from the reader to see. You might magnify the important par. |
||
|
Response Thank you for pointing this out. We used arrows to point the secretion in figure 2.
|
||
|
Comments 12- Kindly add the name of the used software used for the results analysis. |
||
|
Response Thanks for your suggestion, we have added data analysis methods and software into the new subsection 4.6, “Data Analysis”. |
||
|
Comments 13- Kindly include each finding represented in each image. In other words link the many images you have used in the text. |
||
|
Response Thanks for your suggestion, we have updated the manuscript to explicitly connect each key finding with the corresponding figures. Below is a summary of how each image relates to our findings: L66-62 Figure 1 L127-132 Figure 2 L149-151 Figure 3A L157-160 Figure 3B L161 Figure 3C L174-179 Figure 4 L196-202 Figure 5 L211-220 Figure 6 L229-233 Figure 7 L243-245 Figure 8 L257-262 Figure 9 L286-288 Figure 10A L295-297 Figure 10B L300-302 Figure 10C L315-316 Figure 11 L333-335 Figure 12 L413-414 Figure 13 |
||
|
Comments 14- (Optional) Kindly show that the tree did not made the same response if injured by an inert material. |
||
|
Response Thanks for your suggestion, we created MS using carving knives and tweezers to simulate the behavior of ALB females when making oviposition depressions and inserting ovipositors. |
||
|
Comments 15- Describe the name of the used software. |
||
|
Response Thanks for your suggestion, we have added data analysis methods and software into the new subsection 4.6, “Data Analysis”. |
||
|
Comments 16- Add a separate part to the statistical analysis. |
||
|
Response Thanks for your suggestion, we have added data analysis methods and software into the new subsection 4.6, “Data Analysis”. |